# Chasing the Major Sphingolipids on Earth: Automated Annotation of Plant Glycosyl Inositol Phospho Ceramides by Glycolipidomics

**DOI:** 10.3390/metabo10090375

**Published:** 2020-09-19

**Authors:** Lisa Panzenboeck, Nina Troppmair, Sara Schlachter, Gunda Koellensperger, Jürgen Hartler, Evelyn Rampler

**Affiliations:** 1Department of Analytical Chemistry, Faculty of Chemistry, University of Vienna, Waehringer Str. 38, 1090 Vienna, Austria; lisa.panzenboeck@univie.ac.at (L.P.); a01556591@unet.univie.ac.at (N.T.); a11774156@unet.univie.ac.at (S.S.); gunda.koellensperger@univie.ac.at (G.K.); 2Vienna Metabolomics Center (VIME), University of Vienna, Althanstraße 14, 1090 Vienna, Austria; 3Chemistry Meets Microbiology, University of Vienna, Althanstraße 14, 1090 Vienna, Austria; 4Institute of Pharmaceutical Sciences, University of Graz, Universitätsplatz 1/I, 8010 Graz, Austria; juergen.hartler@uni-graz.at

**Keywords:** glycolipidomics, GIPC, glycosyl inositol phospho ceramides, Lipid Data Analyzer, lipidomics, sphingolipids, ultra-high pressure liquid chromatography, high-resolution mass spectrometry, LC-MS, automated annotation

## Abstract

Glycosyl inositol phospho ceramides (GIPCs) are the major sphingolipids on earth, as they account for a considerable fraction of the total lipids in plants and fungi, which in turn represent a large portion of the biomass on earth. Despite their obvious importance, GIPC analysis remains challenging due to the lack of commercial standards and automated annotation software. In this work, we introduce a novel GIPC glycolipidomics workflow based on reversed-phase ultra-high pressure liquid chromatography coupled to high-resolution mass spectrometry. For the first time, automated GIPC assignment was performed using the open-source software Lipid Data Analyzer (LDA), based on platform-independent decision rules. Four different plant samples (salad, spinach, raspberry, and strawberry) were analyzed and the results revealed 64 GIPCs based on accurate mass, characteristic MS2 fragments and matching retention times. Relative quantification using lactosyl ceramide for internal standardization revealed GIPC t18:1/h24:0 as the most abundant species in all plants. Depending on the plant sample, GIPCs contained mainly amine, N-acetylamine or hydroxyl residues. Most GIPCs revealed a Hex-HexA-IPC core and contained a ceramide part with a trihydroxylated t18:0 or a t18:1 long chain base and hydroxylated fatty acid chains ranging from 16 to 26 carbon atoms in length (h16:0–h26:0). Interestingly, four GIPCs containing t18:2 were observed in the raspberry sample, which was not reported so far. The presented workflow supports the characterization of different plant samples by automatic GIPC assignment, potentially leading to the identification of new GIPCs. For the first time, automated high-throughput profiling of these complex glycolipids is possible by liquid chromatography-high-resolution tandem mass spectrometry and subsequent automated glycolipid annotation based on decision rules.

## 1. Introduction

The sphingolipidome of plants contains glycosyl inositol phospho ceramides (GIPCs), glycosylceramides and ceramides, whereas sphingomyelin, globosides, sulfatides or gangliosides are absent. GIPCs were characterized as the major sphingolipid on earth due to their high abundance in plants and fungi, which comprise a large portion of the biomass of the biosphere [1]. GIPCs were first described more than 60 years ago as “phytoglycolipids” [2]. The total plant lipid content can consist of up to 40% GIPCs [3]. The structure of these plant sphingolipids has three major subunits: (1) a polar inositol containing part, (2) the sphingoid backbone with a long-chain base (amino-alcohol) linked by an amide bond to a (3) fatty acyl chain moiety [2,4]. The terms d, t and q refer to the hydroxylation state of the whole ceramide or long-chain base (LCB) moiety, ranging from two (d) to four (q) hydroxy groups. The term h denotes a hydroxylation of the fatty acyl group (i.e., the ceramide moiety q40:1 can correspond to a t18:1 LCB connected to a h22:0 fatty acyl). Di- and trihydroxylation of LCBs with t18:0, t18:1(8Z and 8E) (the main sphingoid base in some species), and d18:0, d18:1(8Z and 8E), d18:2 (4E/8Z and 4E/8E) and fatty acid components varying in chain-length, saturation and hydroxylation state (h16:0–h26:1, 20:0 to 28:0) have been reported in plant GIPCs [5,6]. Different GIPC core structures were determined from higher plants ranging from simple high-abundant A-series species with Hex-HexA-IPC and HexN(Ac)-HexA-IPC (Hex = hexose, HexA = hexuronic acid, IPC = inositol phospho ceramide, HexN = hexosamine, and HexNAc = N-acetyl hexosamine) to low abundant F-series species containing several arabinoses and hexoses [3,7]. Despite the fact that GIPCs are an integral part of the plant plasma membrane, there is still little knowledge concerning its molecular organization and the way this organization is involved in signaling processes necessary for cellular adaptation [1]. To understand the interplay of GIPCs with different enzymes and their detailed function in the plasma membrane in plants, comprehensive structural information provided by observation tools such as NMR or MS are necessary. 

Even though GIPCs were discovered 60 years ago, their analysis remains challenging due to the lack of available standards, automated annotation software and reference databases. For example, CHEBI [8] does not provide any GIPCs and the comprehensive LIPID MAPS Structure Database (LMSD) contains only one GIPC (A-NH_2_-t18:1/h24:0) [9]. As GIPCs consist of a sugar head group linked to a lipid subunit causing amphiphilic properties, they are neither well covered by common glycomics nor lipidomics workflows. Consequently, specialized glycolipidomics analysis strategies are required, e.g., applying a mixture of 2-propanol (IPA), hexane and water [10]. The combination of liquid chromatography and mass spectrometry (LC-MS) has been used due to its unprecedented potential to annotate GIPCs by *m/z*, retention time and fragmentation pattern [7,11]. Unambiguous GIPC identification requires both retention time evaluation and detection of structural subunits by tandem mass spectrometry (MS2), due to the absence of commercial standards. Most GIPC LC-MS-based analysis workflows were performed almost a decade ago by electrospray ionization followed by analysis with low resolution mass spectrometers (QQQ, QTRAP) [7,11]. Meanwhile, high-resolution mass spectrometers (such as TOF, orbitrap, FTICR) have been established with up to 1,000,000 resolution enabling GIPC analysis by accurate mass [12]. Additionally, ultra-high pressure liquid chromatography (up to 1500 bar) with sub 2-µm particles provides high chromatographic resolution and excellent sensitivity. Up to now, GIPC analysis has been performed by tedious manual annotation and curation [1,7,12,13] and expert knowledge was necessary to interpret glycosphingolipid tandem mass spectrometry fragmentation patterns [14,15,16]. The instrumentation advancements of the recent years paved the way for automated high-throughput GIPC analysis. In this work, a variety of plants, i.e., iceberg lettuce (*Lactuca sativa var. capitata nidus tenerimma*), deep frozen spinach (*Spinacia oleracea*), raspberries (*Rubus idaeus*), and strawberries (*Fragaria*) were analyzed by the combination of reversed-phase (RP) ultra-high pressure liquid chromatography (UHPLC) and high-resolution mass spectrometry (HRMS). For the first time, automated GIPC annotation will be performed using the Lipid Data Analyzer (LDA) and platform-independent decision rules [17]. 

## 2. Results

Here we describe a novel workflow by RP-HRMS/MS using the open-source program LDA [17] for automated GIPC assignment. Method development considerations and guidelines for the automated structural analysis of GIPCs are provided. Finally, we test the developed glycolipidomics workflow for different plant samples, leading to a reference database of GIPCs, including fragmentation and retention time information.

### 2.1. Method Development for Automated GIPC Assignment 

GIPCs were extracted by a mixture of IPA, n-hexane and water [18]. So far, most LC-MS-based GIPC chromatographic separations relied on the use of tetrahydrofuran (THF) containing solvents [7,11,12,13]. However, the usage of THF in the eluent system has some drawbacks: (1) it is aprotic and cannot donate a proton; thus, for ionization, pairing with a protic solvent (usually water) is necessary; (2) it can attack tubing (especially PEEK tubing); (3) it tends to polymerize (usually in APCI mode); and (4) it is highly flammable. In order to avoid the use of THF, we developed a novel GIPC method based on RP-HRMS/MS, facilitating a 30 min isopropanol gradient (detailed information can be found in the Materials and Methods Section 4.3). GIPC detection was performed using both negative and positive electrospray ionization and high-resolution Orbitrap MS (see Materials and Methods Section 4.4). Importantly, GIPC analysis requires relatively high RF voltages (S-lens RF level of 45) to ensure efficient transport of medium size glycolipids in the mass spectrometer. Figure 1 shows the extracted ion chromatogram of GIPCs in salad samples analyzed by RP-HRMS, based on data-dependent MS2 (ddMS2) in positive and negative ion modes. The GIPCs displayed in Figure 1 belong to the A-series (Hex(R1)-HexA-IPC) with R1 being a hydroxyl group and the ceramide portion consisting of a hydroxylated saturated fatty acyl chain attached to a t18:1 long chain base. 

As no commercial standards are available, GIPC assignment has to be conducted with caution. In such a situation, the use of the equivalent carbon number model (ECN) is required [19,20]. The ECN model originates from state of the art lipidomics workflows and is based on elution orders observed in RP columns: (1) longer fatty acid chains will increase the retention time (see Figure 1) and (2) more double bonds will decrease the retention time [21] (see Appendix A). To increase the level of confidence in GIPC annotation, we accepted only GIPCs that: (1) were detectable by accurate mass (±5 ppm) in MS1 at the same retention time in both positive and negative ion modes (Figure 1); (2) showed MS2 spectra with characteristic fragments for the ceramide and sugar part in at least one ion mode and; (3) fulfilled the ECN model.

### 2.2. Structural Elucidation and GIPC Annotation Based on MS2 Information

In this work, we introduce the first automated GIPC annotation workflow based on structural information provided by acquired MS2 spectra. Structural analysis and automated GIPC annotation was performed based on a set of in-house developed decision rules for the freely available software LDA [17,22]. As no standards were available, blank extractions (no GIPC annotations found) and GIPC annotations in salad [13] and spinach [12] reported in the literature were used to validate GIPC assignments (Figure 1, Table A1 and Appendix A). Various LCBs (d18:0, d18:1, d18:2, t18:0, and t18:1) and fatty acids (FAs) (16–26) with or without hydroxylation have been reported [5,13]. Moreover, R1 in Figure 2A can either be a hydroxyl (OH), an amine (NH_2_) or an N-acetylamine (NAc) group, increasing the number of putative GIPCs even within a single series.

The final decision rule set was based on well-defined fragments (fragment rules) and their intensity relationships (intensity rules) (Folder S1). The characteristic fragments [IP]^−^ (*m/z* 259) and [IP-H_2_O]^−^ (*m/z* 241) are mandatory in negative ion mode (e.g.: Figure 2B). However, these fragments are not specific, since they are produced by other phosphoinositol-containing lipids too. Thus, for a confident identification, negative or positive ion mode fragments indicating the sugar or ceramide part have to be detected. 

In the majority of cases (see level 2 annotations, Table A1 and Appendix A), MS2 spectra with GIPC fragmentation patterns were detected in both negative and positive mode. Depending on the fragmentation pattern and the level of confidence [24] of the structural elucidation, GIPCs are assigned as either: (1) series-R1-hydroxylation stage-carbon number (LCB + FA)-number of double bonds (LCB + FA) if the exact ceramide composition is not known or (2) series-R1-LCB/FA. Figure 2B displays an exemplary ddMS2 spectrum of A-OH-q42:1 with *m/z* 1260.7237, in salad recorded in negative ion mode. The positive ion mode fragmentation pattern of the [M + H]^+^ precursor (*m/z* 1262.7389, Figure 2C) revealed further structural details, based on the identification of [W]^+^, [W-H_2_O]^+^ and [W-2H_2_O]^+^ fragments, indicating an A-OH-t18:1/h24:0 GIPC. Additional GIPC confirmation is possible by Z_0_ fragments ([Z_0_]^+^, [Z_0_-H_2_O]^+^) of the [M + H]^+^ precursor and by the sodium adduct [M + Na]^+^ (Figure A1), where sugar fragments are readily observable. GIPCs were annotated based on single ionization information only if (1) in negative ion mode in addition to the apparent [IP]^−^/[IP-H_2_O]^−^/[H_2_PO_4_]^−^ fragments at *m/z* 259, 241 and 97, other characteristic fragments were detectable e.g., [C_3_PO_3_]^−^ (*m/z* 596 − R1 = NH_2_, *m/z* 597 − R1 = OH, *m/z* 638 − R1 = NAc), [C_3_PO_3_-C_1_-CO_2_]^−^ (*m/z* 373) or [C_3_PO_3_-C_1_-CO_2_-H_2_O]^−^ (*m/z* 355) or (2) in positive ion mode the [IP]^+^ (*m/z* 261)/[IP + Na]^+^ (*m/z* 283) and fragments indicating the ceramide moiety (e.g., Z_0_) were identified by LDA. The detailed fragment information used for GIPC annotation can be found in Appendix A.

GIPC annotation can be hampered by the presence of isobaric masses for qX:Y NH_2_ and t(X − 2):(Y − 1) NAc (where X refers to the carbon number (LCB + FA) and Y refers to the number of double bonds (LCB + FA), respectively). This may result in false positive GIPC identifications, because these classes share the same characteristic fragments *m/z* 241, 259, 355, 373 and 417. The correct structural elucidation is possible if additional fragments such as [C_3_PO_3_]^−^ (R1 = OH − *m/z* 597, R1 = NH_2_ − *m/z* 596, R1 = NAc − *m/z* 638) in negative ion mode or if LCBs in positive ion mode can be identified based on [W]^+^, [W-H_2_O]^+^ and [W-2H_2_O]^+^ fragments. In the ddMS2 spectra of the [M + H]^+^-precursor, trihydroxylated LCBs are characterized by the presence of three W fragments ([W]^+^, [W-H_2_O]^+^ and [W-2H_2_O]^+^), such as t18:0 (*m/z* 300, 282 and 264) and t18:1 (*m/z* 298, 280, 262), while dihydroxylated species miss the [W-2H_2_O]^+^ fragment, e.g., d18:0 (*m/z* 284, 266), d18:1 (*m/z* 282, 264) and d18:2 (*m/z* 280, 262). As such, both LCB hydroxylation levels can be clearly distinguished. 

### 2.3. Analysis of Different Plant GIPCs by UHPLC-HRMS Suggesting t18:2 LCB 

The novel RP-HRMS/MS and GIPC annotation workflow was used to analyze different plant samples, namely salad (*Lactuca sativa var. capitata nidus tenerimma*), deep frozen spinach (*Spinacia oleracea*), raspberries (*Rubus idaeus*) and strawberries (*Fragaria*). As glycosphingolipid analysis is not negatively impacted by alkaline hydrolysis [10], alkaline hydrolysis was performed to simplify lipid profiles by removing the phospholipid background in the unknown plant samples (strawberry and raspberry, detailed information can be found in the Materials and Methods Section 4.2.2). Figure A2 shows the RP-HRMS/MS GIPC profile for the five most abundant GIPCs determined in spinach, strawberry and raspberry samples. For the sake of clarity, the five most abundant GIPCs in salad (A-NAc-t18:1/h24:0, A-NH_2_-t18:1/h24:0, A-OH-t18:1 h22:0 and h24:0, A-OH-t18:0/h24:0) are not displayed in Figure A2. Irrespective of the plant sample, the species group A-R1-t18:1/h24:0 was always the most abundant one. While in spinach R1 was always N-acetylamine (A-NAc-t18:1 h22:0 to h26:0) for the five dominating GIPCs, in strawberries the major GIPCs contained a hydroxyl group as R1 (A-OH-t18:1 h23:0 to h26:0 and A-OH-t18:0/h24:0). In contrast to that, raspberries had an amine group as R1 for four out of five shown GIPCs (A-NH_2_-t18:0/h24:0, A-NH_2_-t18:1 h22:0 and h24:0, A-NH_2_-t18:2/h24:0 and A-OH-t18:1/h24:0), emphasizing the structural diversity of GIPCs in different plants. 

By analyzing different GIPCs, the NAc, NH_2_ and OH-species from the A series could be detected (Figure 3A–C) with high confidence by (1) accurate determination of mass, (2) matching retention times of ion modes, (3) characteristic fragments and (4) the ECN model. We recommend checking isotopic patterns to avoid false positive hits. For a comprehensive overview of the annotated GIPCs see Table A1.

Due to the absence of commercially available GIPC standards, relative quantification of the individual species was performed using C16 lactosyl(ß) ceramide (d18:1/16:0) as the internal standard. This compound is similar in structure (sugar and ceramide moiety) and retention time (14 min). Even though lactosyl ceramide (d18:1/16:0) may be present in plants, we could not detect it in our samples, thus, making it suitable as the internal standard in our workflow. Normalization by the internal standard (area ratio) and dry weight was performed for MS1-based relative quantification by Skyline [25] (Figure 3A–C). Estimated concentrations in the nmol to µmol range per gram dry weight were observed, which is consistent with the literature [12,18].

In summary, 64 GIPCs in salad (19), spinach (8), strawberry (10) and raspberry (27) were annotated (Table A1). Ranking of the GIPC annotations was performed according to the guidelines of the metabolomics society [24,26], leading to 48 level 2 (matching accurate masses and MS2 in negative and positive mode) GIPCs, 13 level 3 (MS2 in one ion mode with matching accurate masses in both ion modes) GIPCs and 3 level 3** (matching accurate masses in both ion modes, MS2 in one ion mode but lacking information on IP fragments in positive ion mode or lacking sugar information in negative ion mode) GIPCs. The annotations found in spinach and salad are in accordance with literature [12,13]. To the best of our knowledge, this is the first report on GIPCs in strawberries and raspberries. 

Within the annotated GIPCs with structural information on LCB and fatty acyl composition, t18:1 followed by t18:0 and t18:2 were the most prominent LCBs in the analyzed plants (Table A1). For GIPCs containing N-acetylamine residues t18:1 was the most abundant LCB with regard to normalized ratios per gram dry weight (Figure 3A). The same holds true for the amine or hydroxyl group containing GIPCs, with additional high abundance of t18:0 LCBs (Figure 3B,C). While in spinach solely t18:1 LCBs were detected, salad, strawberries and raspberries show more variation in terms of LCB composition with presence of both t18:1 and t18:0 (Table A1). 

Interestingly, besides the expected t18:0 and t18:1 LCBs (R1 = NAc, NH_2_, OH), we additionally annotated four t18:2 (R1 = NAc, NH_2_, OH) species in raspberries (Figure 3A–C). These annotations are verified by coinciding retention times in positive ion mode (Figure A3A), detection of characteristic fragments in MS2 spectra (Figure A3B,C) and conformity with the ECN model (Appendix A). However, we could not find any report in the literature of t18:2 species, which can be explained as up to now no automated GIPC annotation was possible and t18:2 GIPC species were only detected in raspberries. As no standards are available, it is difficult to prove the presence of this species and further investigation is needed. A confirmed t18:2 LCB would indicate a much higher diversity in sphingolipids than anticipated in the past. Another hint for the complex nature of GIPCs in raspberries is the additional annotation of GIPCs with di- and trihydroxylated variants compared to all other analyzed plants. 

Concerning LCB and fatty acyl combinations, t18:1/h22:0 and t18:1/h24:0 showed equal annotation numbers for N-acetylamine or amine containing GIPCs (Figure 3A,B). Independent of the NH_2_, OH or NAc functional group, only two odd chain fatty acids (h23:0, h25:0) were detected and no fatty acids with a length from 17 to 21 carbon atoms were found. For the hydroxyl group containing GIPC variants, the combinations t18:1/h22:0, t18:1/h23:0, t18:0/h24:0, t18:1/h24:0 and t18:1/h25:0 were found in equal annotation numbers. Overall, plant GIPCs with a combination of t18:1 LCB and a h24:0 fatty acyl moiety were the most abundant ones in terms of normalized ratios per gram dry weight (Figure 3A–C, Table A1).

## 3. Discussion

GIPCs are the major sphingolipids on earth [1]. Hence, it is important to understand their function and distribution in plants and fungi. However, GIPC analysis remains extremely challenging, as tailored extraction strategies for this glycolipid class are necessary. GIPC analysis is in its infancy due to the lack of standards and databases. In this work, we present the first automated high-throughput GIPC annotation workflow which is based on RP-HRMS/MS. By using a novel 30 min gradient based on isopropanol with a reversed-phase column, packed with sub 2-µm particles, fast GIPC analysis was possible at the same time avoiding standard eluent use of tetrahydrofuran. Four different plant samples were analyzed. For salad and spinach, literature information has been available [12,13], while for raspberry and strawberry, GIPC profiles were completely uncharacterized. Using strict filtering by (1) accurate mass determination (±5 ppm) with matching retention times for both ion modes in MS1, (2) MS2 spectra with characteristic fragments and (3) expected retention time series, we produced a database of 64 GIPCs (Table A1). As no GIPC standards are available, only GIPC annotation hits with level 2 and 3 confidence [24] were possible. The most prominent MS2 fragments for GIPCs are [IP] fragments in both ion modes ([H]^−^: *m/z* 241, 259; [H]^+^: *m/z* 261; [Na]^+^: *m/z* 283). However, additional sugar or ceramide fragments are essential for correct GIPC annotation. The high MS2 mass range coverage (*m/z* 65 to 2500) provided by the Orbitrap was beneficial to determine GIPC low mass fragments such as *m/z* 79 [PO_3_]^−^ or 97 [H_2_PO_4_]^−^, besides high mass precursors such as 1261 [M − H]^−^ (Figure 2).

Relative quantification with the internal standard lactosyl ceramide revealed GIPC t18:1/h24:0 as the most abundant species, independent of the plant sample. Depending on the plant sample, GIPCs contained mainly amine, N-acetyl or hydroxyl residues. Most GIPCs showed a Hex-HexA-IPC core with a trihydroxylated t18:0 or t18:1 long-chain base ceramide part a and hydroxylated fatty acid chains ranging from h16:0 to h26:0. Interestingly, in raspberry, four GIPCs contained t18:2, which was not reported so far. This finding would suggest the existence of more complex sphingolipid species in nature than previously anticipated. Further analysis by orthogonal methods such as NMR, GC-MS or IMS and available GIPC standards would be necessary to confirm the presence of the t18:2 GIPC group. Different analytical strategies could also resolve potential isomeric species and provide comprehensive details on the sugar moiety present in GIPCs. Nevertheless, this example shows the power of this workflow to detect promising novel GIPC candidates in an automated fashion. In order to support LC-MS-based GIPC analysis in general, we provide the mass lists for GIPCs in positive and negative ion modes (Appendix A), as well as the fragmentation rules (Folder S1) for setting up the automated GIPC analysis by Lipid Data Analyzer. Even though we confirmed the GIPCs exclusively from the A-series, the presented strategy is also suitable to determine less or more complex GIPC series, such as 0, B, C, D, E and F. However, extended analytical workflows (e.g., multi-stage fragmentation/MSn) and additional software method development might be necessary. Precursor mass lists for positive ([M + H]^+^) and negative ([M − H]^−^) ion modes comprising series 0–F, LCBs d18:0, d18:1, d18:2, t18:0 and t18:1 and fatty acyls h15:0–h26:0, h15:1–h26:1 and n20:0–n28:0 (n = non-hydroxylated), as reported in the literature [5,13], can be found in Appendix A. In general, we believe that LC-HRMS/MSn combined with automated annotation based on decision rules will pave the way for more complex glycolipidomics profiling. 

## 4. Materials and Methods 

### 4.1. Material

The plant material used was derived from salad (*Lactuca sativa var. capitata nidus tenerimma*), deep frozen spinach (*Spinacia oleracea*), raspberries (*Rubus idaeus*) and strawberries (*Fragaria*). (A more detailed description of plant samples can be found in Table A2.)

All chemicals were of LC-MS grade. Acetonitrile (ACN), methanol (MeOH), IPA and water were bought from Honeywell (Offenbach, Germany) and n-hexane was bought from VWR (Vienna, Austria). Butylated hydroxytoluene (BHT) was purchased from Sigma-Aldrich (Vienna, Austria), ammonium formate (AF) from Sigma-Aldrich (Vienna, Austria) and formic acid from VWR (Vienna, Austria). C16 Lactosyl(ß) Ceramide (d18:1/16:0) (D-lactosyl-ß-1,1’ N-palmitoyl-D-erythro-sphingosine) was purchased from Avanti Polar Lipids, Inc. (Alabaster, Albama, USA), was used as internal standard (IS) and dissolved in an appropriate amount of IPA to achieve a concentration of 100 µM. 

### 4.2. Sample Preparation

Salad was manually cut into small pieces before being weighed into falcon tubes (50 mL, VWR, Vienna, Austria) using a CPA225D balance (Sartorius, Vienna, Austria). Raspberries and strawberries (whole fruits) were homogenized with a hand blender (Tefal/SEB, Ecully, France). Raspberries, strawberries and deep-frozen homogenized spinach were directly weighed into 10 mL glass vials (more details can be found in Table A2). In order to prevent potential oxidation of lipids, 3 mL of an approximately 0.01% BHT solution in IPA were added and samples were mixed. Subsequently 30 µL IS were spiked into all samples except for one replicate (to test for potential IS presence in plants). Salad samples were homogenized using an ultra-turax (miccra d-1, Heitersheim, Germany) which was cleaned with 70% IPA and dried between the samples. In order to inhibit lipase activity, all samples were incubated at 75 °C for 30 min under constant shaking [27]. The warm salad samples were subsequently transferred into glass vials. The following sections provide a detailed overview of the extraction strategies that were applied.

#### 4.2.1. One-Phase Extraction

The extraction of GIPCs from salad and spinach was performed as previously reported [18] using a mixture of IPA, n-hexane and water. Amounts of 3.47 mL IPA, 0.6 mL n-hexane and 1.93 mL water were added to the salad and spinach samples. In order to ensure sufficient accessibility of the plant material, samples were vortexed and manually shaken prior to incubation at 60 °C for 15 min under constant shaking. 

#### 4.2.2. One-Phase Extraction Combined with Alkaline Hydrolysis

To avoid the occurrence of glycerophospholipids, which might reduce GIPC ionization efficiency and lead to potential false identifications, alkaline hydrolysis was applied for the raspberry and strawberry samples, using an adapted workflow [28]. After incubating the plant material with the BHT solution for 30 min at 75 °C under constant shaking, 3.47 mL IPA and 0.6 mL n-hexane were added. Samples were vortexed and put on a shaker for 15 min at 60 °C. As soon as the samples had reached room temperature 707 µL 1 M KOH in MeOH was added and the solution was vortexed. After shaking the samples for 2 h at 37 °C, they were left at room temperature. Subsequently 100% formic acid was added until a pH of ~6–7 was reached and 1.93 mL water was added before repeating the incubation step. 

#### 4.2.3. Centrifugation, Drying and Reconstitution

Irrespective of the extraction strategy, the warm samples were centrifuged at 1000 rpm for 10 min at 4 °C and the supernatant was transferred into a separate glass vial. The solvent was evaporated to dryness overnight in a Genevac EZ-2 Series Personal Evaporator (SP Scientific, Ipswich, UK) and the dried residue was reconstituted in 2 mL IPA:H_2_O (65:35) [13]. Samples were vortexed prior and after ultrasonication at 30 °C for 15 min. Subsequently, 500 µL of this solution was filtered directly into HPLC vials through a ClariStep filter (Sartorius, Vienna, Austria). Pools were prepared separately for each plant by pipetting 50 µL of each biological replicate into a separate HPLC vial. A quality control pool was prepared by combining 30 µL of the pooled samples.

### 4.3. Reversed-Phase Chromatography

Liquid chromatography was performed using a C18 Acquity UHPLC HSS T3 reversed phase column (2.1 × 150 mm, 100 Å, 1.8 µm, Waters, Vienna, Austria) equipped with a VanGuard Pre-column (2.1 × 5 mm, 100 Å, 1.8 µm, Waters, Vienna, Austria) at a column temperature of 40 °C. The flow rate was 0.25 mL/min and the backpressure was 460 bar at the starting conditions. Gradient elution with a total runtime of 30 min was performed using the solvent A: ACN:H_2_O (3:2, *v/v*) and the solvent B: IPA:ACN (9:1, *v/v*), both of which contained 0.1% formic acid and 10 mM ammonium formate. 

The gradient can be described as follows: 0–2 min 30% B, 2–3 min ramp to 55% B, 3–17 min ramp to 67% B, 17–22 min ramp to 100% B, 22–26 min 100% B, followed by an equilibration step from 26 to 30 min using 30% B. A Vanquish Duo UHPLC system (Thermo Fisher Scientific, Germering, Germany) was used and injections were performed with an autosampler. An injection volume of 10 µL was chosen and the injector needle was flushed with 75% IPA and 1% formic acid in between the injections.

### 4.4. High-Resolution Mass Spectrometry

The LC system was coupled to a Q Exactive HF (Thermo Fisher Scientific, Bremen, Germany) high resolution mass spectrometer, applying a HESI ion source with an S-lens RF level of 45. Measurements were carried out in positive and negative modes using different parameters. The following settings were applied in positive mode: spray voltage: 3.5 kV, capillary temperature 220 °C, sheath gas flow rate: 30, and auxiliary flow rate: 5. In negative mode parameters were adapted as follows: spray voltage: 2.8 kV, capillary temperature 250 °C, sheath gas flow rate: 35 (a.u.), and auxiliary flow rate: 10 (a.u.). The top 10 data-dependent MS2 spectra were obtained at a scan range of 500 to 3000 *m/z* with HCD using normalized collision energies of 35 (+35 in positive ion mode, −35 in negative mode), an MS1 resolution of 15,000 or 30,000 with an AGC target of 1e6 and MS2 resolution of 15,000 with an AGC target of 1e5. MS2 spectra were acquired based on an inclusion list (“do not pick others” option) containing the GIPC series 0–F (*m/z* values were calculated using enviPat Web 2.4 [29]). A more comprehensive picture of the GIPC composition of the analyzed plant material was obtained using several rounds of automatically generated exclusions lists for the sample pools [30].

### 4.5. Data Analysis

The GIPC assignment was performed using LDA (version 2.8.0) [17]; corresponding settings (Table A3), mass lists (Appendix A) and decision rule sets for series A (Folder S1) can be found in the Appendix B and Appendix A. The correct GIPC annotation was ensured by a manual inspection of the results. MS1-based relative quantification of annotated GIPCs was performed with Skyline [25]. Total areas were divided by the corresponding calculated dry weights and areas of the IS, resulting in normalized ratios per g dry weight, of which the average was taken based on the number of replicates (3 for salad and spinach, 4 for strawberries and raspberries). More information can be found in Appendix C.

## Figures and Tables

**Figure 1 metabolites-10-00375-f001:**
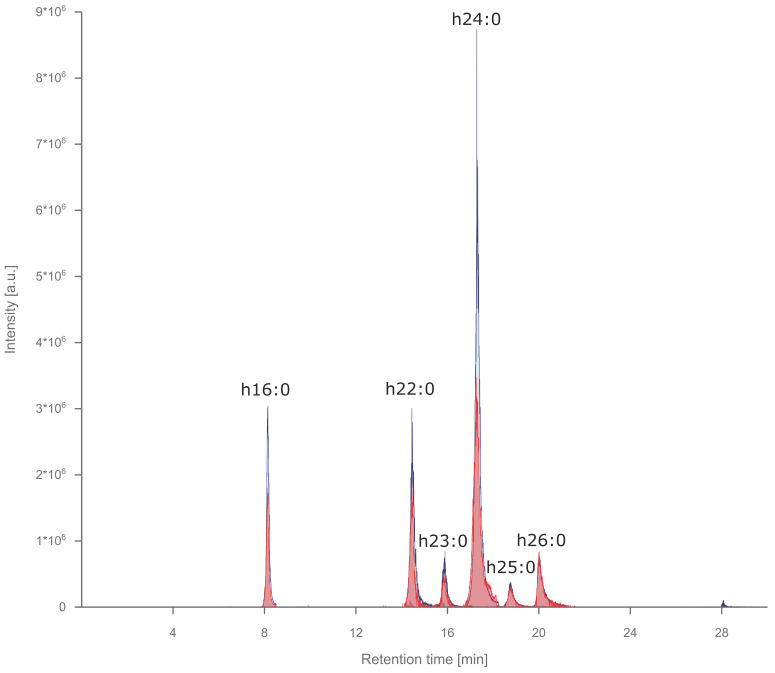
Extracted ion chromatogram of glycosyl inositol phospho ceramides (GIPCs) in salad samples analyzed by RP-HRMS/MS analysis using ddMS2 in positive (red) and negative (blue) ion modes. Assigned GIPCs belong to the Hex-HexA-IPC series with a t18:1 long-chain base (LCB) and varying chain length of the hydroxylated saturated fatty acids. Retention times coincided in positive and negative ion modes. Increasing carbon numbers result in belated elution.

**Figure 2 metabolites-10-00375-f002:**
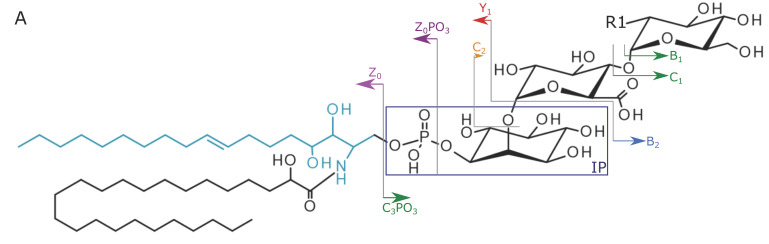
Overview of the GIPC fragmentation for the example of GIPC A-OH-t18:1/h24:0 in salad: (**A**) The fragment assignment of GIPC A-OH-t18:1/h24:0 (adapted from [23]). The W fragment is shown in a light blue color. Please note that a full structural characterization is not possible by RP-HRMS/MS, (**B**) The product ion spectrum in negative ion mode at *m*/*z* 1260.7237, showing characteristic fragments *m/z* 241 and 259, 355, 373 and 417. The sugar head group was confirmed by the [C_3_PO_3_]^−^ fragment (*m/z* 597, R1 = OH). [Z_0_PO_3_]^−^ and [Y_1_-H]^−^ fragments prove the ceramide moiety. (**C**) The positive ion mode ddMS2 spectrum of the [M + H]^+^ precursor, exhibiting the [W]^+^, [W-H_2_O]^+^ and [W-2H_2_O]^+^ fragments at *m*/*z* 298, 280 and 262, which are characteristic for the t18:1 LCB.

**Figure 3 metabolites-10-00375-f003:**
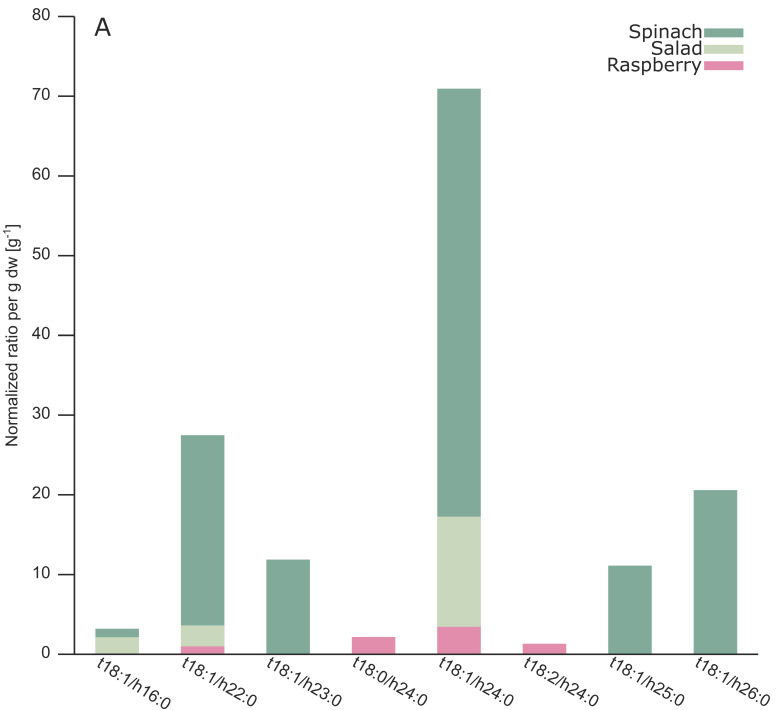
The normalized ratio per gram dry weight for annotated GIPCs in salad (light-green), spinach (green), strawberries (rose) and raspberries (dark-red), by using different substituents for the functional group R1: (**A**) NAc, (**B**) NH_2_, and (**C**) OH (more detailed information can be found in Table A1).

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
