# Peer review of "Chasing the Major Sphingolipids on Earth: Automated Annotation of Plant Glycosyl Inositol Phospho Ceramides by Glycolipidomics"

_metabolites, 2020, doi:10.3390/metabo10090375_

Round 1

Reviewer 1 Report

The authors present a very convincing piece of work.

I would like to ask for some minor addition as the authors describe potential new structures for instance in raspberry. It should be noted that until now structural elucidation of complex glycosphingolipids should be supported by NMR and component analysis by GC-MS. Connectivities between sugars etc. are not determined with the presented method. Furthermore, retention time overlap for isobaric structures, even with application of high mass accuracy does not prove the right assignment of structures, even MS2 would not be sufficient.

I would also like to ask to add in discussion or introduction a short paragraph on the usage of ion mobility MS to assign complex sugars in view of connectivity analysis.

In summary, the study of Panzenboeck et al. can be accepted for publication after some minor addition to the draft.

Reviewer 2 Report

The manuscript entitled “Chasing the major sphingolipids on earth: Automated annotation of plant glycosyl inositol phospho ceramides by glycolipidomics” by Lisa Panzenboeck et al. describes the analysis of glycosyl inositol phospho ceramides (GIPCs) from different plants by use of an omics approach with C18 HPLC-MS2, LDA, and home-made decision rules finally leading to a first GIPC reference database. The experiments seem to be performed carefully and the results are sound. However, the reviewer has a few concerns as outlined below which should be addressed in a minor revision.

The terms “positive ionization” and “negative ionization” should be replaced by “positive ion mode” and “negative ion mode” throughout the manuscript (an ionization is neither positive nor negative).

Fig. 1 neither shows “the developed GIPC RP-HRMS/MS assay” nor the “RP-HRMS/MS analysis of GPICs”. It shows extracted ion chromatograms. The underlying fragment ions for positive and negative ion mode should be given. The unit at the intensity axis is missing.

Fig. 2B: Since the authors claim to determine “accurate masses” (± 5 ppm) they should comment on the deviations of [M−H]: Δ = −14.7 mU ≡ 11.7 ppm and Y1: Δ = 17 mU ≡ 18.4 ppm, while all other m/z values are within the given range. According to the fragment ion nomenclature of Domon and Costello in negative ion mode there is no need for labeling the fragment ions with “−H” and the charge; Y1−H could be considered as doubly charged.

Fig. 2C: Z0: Δ = −9 mU ≡ 14.1 ppm? Correspondingly, the fragment ions in positive ion mode should not be labeled with an additional proton and the charge.

Except for the negative ion MS2 spectrum (Fig. 2B) no precursor signal and only rather small fragment ions are detectable, even in the MS2 of the sodiated species (Fig. A1) which should require higher collision energies (CEs) for fragmentation. This raises the suspicion that a CE of 35? is rather high. Lowering the CE might lead to larger fragments, thus increasing the information on the structures of GIPCs. The authors should comment on this or, even better, run their experiments once with lower CEs.

As to the t18:2 species (line 234 ff.): are there no corresponding W series fragment ions at m/z 296, 278, and 260 detectable? Otherwise the additional double bond could as well be in the acyl chain.

Chapter 4.4. (line 341 ff.): neither for the S-lens RF level nor for sheath gas flow rates and auxiliary flow rates units are given. What does HCD 35 mean (eV?)?

Line 262: typo: lactosyl instead of lacotsyl

Reviewer 3 Report

This interesting study describes an efficient tool in modern lipidomics. Qualitative and quantitative evaluation of plant sphingolipids is challenging and essential task due to their chemical diversity. The methods are described in details and clear enough to be reproduced. The paper can be accepted in Metabolites Journal after minor revision.

Few points:

  • It would be great to describe in more details in the introduction section how the knowledge about glycosylceramide composition and structure in exact plants can be useful for different applications;
  • The drawback #4 sounds little strange because hexane is even more flammable in comparison with tetrahydrofuran;
  • Since there are no standards for other lipopolysaccharides, this uncertainty casts doubt on the reliability of the entire analysis. Please, provide specific chromatograms and/or other raw data to make readers more confident about newly discovered ceramides in raspberries with t18:2 sphingoid base;
  • Finally, there is a question that this research is easy able to answer. Are all combinations of available in cells long-chain bases and fatty acids equally presented or there are some disproportion? This knowledge can shade light on specificity of ceramide synthases and their potential involvement in more complex processes.

Reviewer 4 Report

In the manuscript entitled “Chasing the major sphingolipids on earth: Automated annotation of plant glycosyl inositol phospho ceramides by glycolipidomics,” the authors developed an automated GIPC annotation workflow via RP-HRMS/MS. The authors developed a RP-HRMS/MS method to analyze GIPCs with a gradient based on isopropanol instead of tetrahydrofuran, which can reduce MS sensitivity (among other disadvantages). The authors used a stringent approach to determine the ECN model as they introduced the first GIPC annotation workflow based on structural information provided by acquired MS2 spectra and retention times. GIPC assignment was performed, for the first time, using Lipid Data Analyzer, which is based on platform-independent decision rules. The authors also appear to be the first to report GIPCs in strawberries and raspberries. This approach has the potential to be used for complex glycolipidomics profiling applications.

Comments:

  1. There should be a flow chart or schematic diagram describing the automated workflow.
  2. The authors should clarify the part of the automated GIPC annotation workflow via RP-HRMS/MS workflow that is high throughput. The LC-MS method does not appear to be high-throughput (or is this relative to other GIPC methods?).
  3. Line 21. Consider revising this sentence – e.g., to “Data from four different plant samples…”.
  4. Line 64. Do you mean “unprecedented potential”?
  5. Line 191. Change to “and raspberry”.
  6. Figure 3. Error bars should be added to the stacked bar graphs.
